# Brief communication: The hidden labyrinth: Deep groundwater in Wright Valley, Antarctica

Hilary A. Dugan[1], Peter T. Doran[2], Denys Grombacher[3], Esben Auken[3], Thue Bording[3], Nikolaj Foged[3], Neil Foley[4], Jill Mikucki[5], Ross A. Virginia[6], and Slawek Tulaczyk[7]

[1]Center for Limnology, University of Wisconsin-Madison, Madison, WI 53705, USA
[2]Department of Geology and Geophysics, Louisiana State University, Baton Rouge, LA 70802, USA
[3]Department of Geoscience, Aarhus University, 8000 Aarhus, Denmark
[4]Environmental Sciences Department, University of Montana Western, MO 59725, USA
[5]Department of Microbiology, University of Tennessee, Knoxville, TN 37996, USA
[6]Department of Environmental Studies, Dartmouth College, Hanover, NH 03755, USA
[7]Department of Earth and Planetary Sciences, University of California, Santa Cruz, Santa Cruz, CA 95064, USA

**Correspondence:** Hilary Dugan (hdugan@wisc.edu)

**Abstract.** Since the 1960s, a deep groundwater system in Wright Valley, Antarctica, has been the hypothesized source of brines to hypersaline Don Juan Pond and Lake Vanda, both of which are rich in calcium and chloride. Modeling studies do not support other possible mechanisms, such as evaporative processes, that could have led to the current suite of ions present in both waterbodies. In 2011 and 2018, an airborne electromagnetic survey was flown over the Wright Valley to map subsurface resistivity (down to 600 m) in exploration of liquid water. The surveys revealed widespread unfrozen brine in the subsurface near Lake Vanda, Don Juan Pond, and in the North Fork of Wright Valley. While our geophysical survey can neither confirm nor deny deep groundwater connectivity between Lake Vanda and Don Juan Pond, it does point to the potential for deep valley-wide brine, likely within the Ferrar Dolerite formation.

## 1 Introduction

Wright Valley in the McMurdo Dry Valleys of East Antarctica is notable for being home to the Onyx River, the longest river in Antarctica; Lake Vanda, the deepest and warmest lake in Antarctica; and Don Juan Pond, one of the saltiest bodies of water on Earth. It is also notable for having a long-standing debate as to the origin and connectivity of groundwater brines in the valley. Both Don Juan Pond and Lake Vanda hold salty calcium-chloride rich brines, distinct from the sodium-chloride brines seen to the south in Taylor Valley (e.g., Blood Falls, Mikucki et al. 2009) and to the north in Victoria Valley (e.g., Lake Vida, Dugan et al. 2014).

The earliest limnological investigations of Wright Valley in the 1960s revealed saline waters in both closed-basin Don Juan Pond and the bottom waters of closed-basin Lake Vanda (Wilson and Wellman, 1962; Armitage and House, 1962; Meyer et al., 1962), but it was not until the summers of 1973-74 and 1974-75 with the undertaking of the Dry Valleys Drilling Project (DVDP) that subsurface hydrology was able to be investigated. Importantly for our investigation were three boreholes.

DVDP4A was drilled in the center of Lake Vanda, DVDP13 was drilled immediately to the west of Don Juan Pond, and DVDP14 was drilled, at the time, to the west of Lake Vanda in the North Fork of Wright Valley (it is now underwater).

In 1973-74, DVDP4A penetrated 17.4 m beneath the sediment/water interface of the bottom of Lake Vanda. The core consisted of 3.5 m of lacustrine sediments, underlain by 2.2 m of glacial sediments, 6.7 m of marine sediments, and eventually basement rock (Cartwright et al., 1974). Cartwright et al. (1974) hypothesized the presence of a groundwater system flowing from Lake Vanda to the west. They later invalidated this hypothesis based on fluid potentials measured in the sediments. They found that there was no change in total head between the lake bottom and the underlying sand and gravel, indicating that groundwater was flowing laterally at the borehole location, and if there was an area of groundwater discharge or recharge with the lake, the borehole did not intersect it (Cartwright and Harris, 1981).

During the 1974-75 summer, DVDP14 was drilled 1.5 km west of Lake Vanda. Total penetration depth was 78 m, with basement rock reached at 27.94 m depth (Mudrey Jr. et al., 1975). All water and sediments in the borehole were solidly frozen, which, at the time, was thought to negate the previous summer's hypothesis that water flowed westward from Lake Vanda into the North Fork. Water was later squeezed from the 28 m of sediments and found to be hypersaline (50-180 ppt) (McGinnis et al., 1981).

Also during the 1974-75 summer, at Don Juan Pond, DVDP13 was drilled 75 m deep, reaching highly fractured basement rock at 12.7 m depth (Chapman-Smith, 1975). Water was encountered in the borehole, and consistently rose to a level 0.65-0.80 m higher than the surface of the pond, which indicated the pond was receiving groundwater discharge. The borehole water had a salinity of 200-225 ppt (McGinnis et al., 1981; Harris, 1981), which is less than the salinity of Don Juan Pond. In Dec 1978, the borehole was revisited. Once cleared of ice, -15.5°C water flowed from the hole for three days as an artesian well before being shut off (McGinnis, 1979). It was hypothesized that the fractured and highly mineralized Ferrar Dolerite might be a confined aquifer, sourcing water from the base of the Antarctic Ice Sheet (McGinnis, 1979; Harris, 1981). The water levels and salinity of the pond vary as well, which suggests a hydrological driver beyond surface conditions (Harris et al., 1979). Geophysical resistivity logs of the DVDP13 borehole showed higher resistivity in the Ferrar Dolerite from 23-33 m depth (McGinnis et al., 1981). McGinnis suggested that this may be a zone of high permeability and high flow, and therefore have less diffusion of salts.

Since the DVDP era, a number of scientists have investigated the origin of salts in the lacustrine systems of the Dry Valleys (Green and Lyons, 2009; Lyons et al., 2005). The oddity of calcium-chloride brines in Wright Valley has led to a special focus on Don Juan Pond. Remote sensing of the surface surrounding Don Juan Pond and laboratory experiments with salts have shown that brines can be generated at the surface via deliquescence in $CaCl_2$ rich sediments (Gough et al., 2017). Surface discharge into the pond has been shown to be correlated with meteorological conditions, implying that surface hydrology plays a role in the water balance of the pond (Dickson et al., 2013). However, most mixing models predict that the salts in Lake Vanda and Don Juan Pond could not have originated solely from surface inputs from the Onyx River or surrounding catchment, and must be derived in part, from a deep groundwater source (Green and Canfield, 1984; Green and Lyons, 2009; Carlson et al., 1990). Furthermore, if a regional groundwater system did exist that connected both waterbodies, water should recharge from higher elevation Don Juan Pond (113 m asl) and discharge to lower elevation Lake Vanda (90 m asl).

Evidence for deep groundwater flow in Wright valley is presented by Toner et al. (2017), who developed a model of brine evolution in Don Juan Pond and found that the ionic ratios in Don Juan Pond could only be explained by the upwelling and evaporation of deep groundwater, and that the turnover time must be relatively short (<1 year). They concluded the presence of a regional groundwater flow system, but note their investigation is equivocal on whether a groundwater connection exists between Don Juan Pond and Lake Vanda. Lake Vanda has much higher Mg:Cl and K:Cl ratios than Don Juan Pond, which suggests

the lake undergoes closed-basin evaporation without a significant outflow, unlike Don Juan Pond where they hypothesize groundwater flows both into and out of the pond (Toner et al., 2017).

     The aforementioned studies in Wright Valley were limited in their spatial observations by the difficulty in carrying out investigations in a remote and protected area. This study provides the first integrative overview of subsurface brines in Wright Valley using non-destructive geophysical measurements. Our research goals were to map water distribution and hydrological

connectivity in order to further our understanding of permafrost hydrogeology.

## 2   Wright Valley AEM survey

Here, we present data from airborne electromagnetic (AEM) surveys flown in Wright Valley in 2011 and 2018 (Foley et al., 2015). Previous examinations of these data in Victoria Valley and Taylor Valley have revealed both the existence of confined brine pockets (e.g., under Lake Vida, Dugan et al. 2015), and expansive groundwater connectivity (e.g., lower Taylor Valley,

Mikucki et al. 2015; Foley et al. 2020), and have transformed our understanding of hydrological connectivity in this polar desert ecosystem.

     Comprehensive methodology on our 2018 AEM survey can be found in Grombacher et al. (2021). Briefly, a transient electromagnetic (TEM) measurement involves a current in a transmitter loop being turned on and off, which generates a time-varying magnetic field in the subsurface, subsequently leading to electrical eddy currents in the subsurface that generate a

secondary time-varying magnetic field. These secondary fields are measured by a receiver coil. Highly conductive regions produce larger amplitude, slowly decaying eddy currents, whereas highly resistive regions give rise to smaller amplitude, quickly decaying currents. Our AEM surveys were conducted by suspending a TEM system beneath a helicopter. Inversion is used to estimate the underlying electrical resistivity structures consistent with observed TEM signals. Here, standard 1D TEM forward modelling approaches are employed (Auken et al., 2015), except in the case of Don Juan Pond where a more complex

forward response is required to account for strong induced polarization effects (Fiandaca et al., 2018). In 2017, a ground-based configuration was used to investigate sites prior to the airborne survey. The ground-based TEM had larger transmitter coils and stacking times, and therefore could penetrate more deeply with potentially higher accuracy. A comparison of the ground-based and airborne TEM data revealed high consistency between the methods (Madsen et al., 2022).

     We focus on three E-W flight lines flown in 2018: Line 1, along the North Fork and Lake Vanda; Line 2, along the South

Fork over Don Juan Pond; and Line 3, along the southern edge of Lake Vanda (Fig. 1a). These flight lines were picked to align with the lowest elevation tracks of Wright Valley, where one might assume brine would pool (although we acknowledge surface water and groundwater flow potential can be very different). Additional E-W flight lines along higher elevation tracks returned

higher resistivity readings, which confirm that the low elevation tracks (Lines 1-3) are the most hydrologically interesting. Additional flight lines were flown N-S across Lake Vanda, but many appear to display 3D effects where measured signals are sensitive to features not beneath the sensor (i.e. to the side/front/or rear). In these cases, forward modelling that employs a 1D assumption may introduce spurious features in an attempt to attribute observed signals to structures directly beneath the sensor. These spurious features, that sometimes manifest as "pantlegs" where a shallow feature is stretched deeper to the sides, can render interpretation of connectivity difficult.

Inversion results are presented as cross-sectional images of resistivity ($\Omega$-m) to infer subsurface properties (i.e. highly resistive ice or bedrock, or highly conductive water and brines). The cross-track sampling width is on the order of 60-70 m wide at the shallowest depths and extends to several hundred meters wide at the bottom of the profiles. The depth of investigation (DOI), where the signal to noise ratio is too low for accurate interpretation, was several hundreds of meters in many areas. In addition, a gap between or within a flight line indicates an area where no earth signal could be reliably detected that exceeded instrumental noise levels (e.g. within Line 1, and between Line 2 and 3). This does not negate the presence of a deeper conductive unit, only that it is not present within the DOI of the employed AEM system.

## 3 Wright Valley AEM findings

The returns from the three along-valley flight lines are intriguing (Fig. 1b-d). Firstly, the surface conductors of Lake Vanda and Don Juan Pond are clear, as is the surface of saline Don Quixote pond at the very western edge of Line 1 (Englert et al., 2014). The top 20 m of Lake Vanda has a resistivity of $\sim$20 $\Omega$-m (a typical value for freshwater). At 60 m depth, resistivity drops to 3.9 $\Omega$-m, and further drops to 0.2 $\Omega$-m at 70 m depth. This is consistent with electrical conductivity (EC) of lake water in Vanda, which shows low EC (< 5 mS cm$^{-1}$) from the surface to 63.6 m depth (in 2014), followed by a rapid increase to 120 mS cm$^{-1}$ between the depths of 63.6 and 78 m (Castendyk et al., 2016). The minimum resistivity of 0.14 $\Omega$-m is recorded from 80-92 m, a depth within the lake sediments that overlaps with the highly saline water (> than lake water) found in the sediments of DVDP4A.

The surface of Don Juan Pond, from 0-4 m, has a resistivity of 1 $\Omega$-m in Line 2. This might appear high for the second saltiest body of water on Earth, but the pond itself is only 20-30 cm deep, and the AEM inversions are averaged over the top 4 m. The averaged resistivity would take into account the porosity of the sediments as well as the conductivity of the groundwater. The conductivity of $CaCl_2$ is greater than the conductivity of an equivalent mass of NaCl, and the relationship between concentration (ppt) and specific conductance (mS cm$^{-1}$), is parabolic, with a maximum SpC of approx. 200 mS cm$^{-1}$ at 300 g $CaCl_2$ L$^{-1}$ (Sun and Newman, 1970).

The second depth of low resistivites (12-14 $\Omega$-m) is seen from 23-36 m. This depth overlaps with the hypothesized aquifer in the Ferrar Dolerite (Harris, 1981). A a recent analyses of soluble salts surrounding Don Juan Pond showed that the salts in hillslope surface wet streaks were most likely derived from Ferrar Dolerite bedrock outcrops based on the chemical signatures of Cl$^-$ and NO$_3^-$ (Toner et al., 2022). This is yet another line of evidence that supports a connection between Ferrar Dolerite and Don Juan Pond brine. Based on barometric efficiency calculations, the Ferrar Dolerite is thought to be highly fractured

with a porosity upwards of 20% (Harris, 1981), which could hold a significant volume of brine. Line 2 also reveals a vertical low resistivity feature approximately 2 km to the east of Don Juan Pond. This vertical feature (possibly a dolerite dyke) would intersect a surface basin which holds VXE-6 pond. This pond is known to hold shallow saline groundwater, and be chemically distinct from other South Fork ponds (Harris, 1981; Toner et al., 2022).

In the Don Juan Pond basin, AEM data provide evidence for regional connectivity. From the Line 2 transect, there is subsurface brine present on either side of the pond, and possibly a connection between VXE-6 and Don Juan Pond. The elevation of the brine appears to be higher than the elevation of the pond. This topographic control could be responsible for the artesian well discharge observed during the DVDP drilling.

From our AEM survey, any valley-wide groundwater connectivity is obscure. To the east of Lake Vanda, there is little evidence of subsurface conductors and it is unlikely there is deep groundwater. From Lake Vanda to the west, there is no obvious shallow (<100 m) connectivity between the lake and the North or South Forks. There is evidence for a layer of shallow brines in the North Fork, possibly a remnant of paleolakes. DVDP14 did not penetrate the permafrost, and it may be that it did not extend deep enough, or the spatial heterogeneity is such that it did not intersect an unfrozen brine layer. The region between Don Juan Pond and Lake Vanda was surveyed by both airborne and ground-based TEM. Neither survey was able to record a signal that exceeded the noise floor of the system in the area to the SW of line 3 (see area of *no signal* on Figure 1a). If any brine is present here, it would be > 500 m deep. If the key to finding subsurface brine in the Wright Valley is to "follow the dolerite", more knowledge is needed of subsurface geology in Wright Valley. The dolerite sills generally have a gentle westward dip (McKelvey and Webb, 1962), but evidence of brine-saturated dolerite is not present on the eastern end of the South Fork transect. If there is a dolerite conduit between the two surface waterbodies, it does not follow the surface topography. To the west of Don Juan Pond at higher elevations along the rock glacier, our airborne and ground-based surveys were unable to measure a signal exceeding the noise floor of the system.. From this, we cannot draw any conclusions about the source of the brine or possible connections to the East Antarctic Ice Sheet.

## 4 Conclusions

From Lake Vanda towards Don Juan Pond in the South Fork, we cannot establish a direct subsurface connection from our data, however, a deep (>500 m) connection or a connection outside our study transects may be present. Certainly, resistivities <100 $\Omega$-m to either side of Don Juan Pond support this speculation. This depth is in the range at which liquid brines were inferred in Taylor Valley from AEM (Mikucki et al., 2015), and also similar to the frozen-unfrozen transition at 183 m found in DVDP10 at New Harbour, Taylor Valley and the inferred frozen-unfrozen transition at 248 m in DVDP11 in front of Commonwealth Glacier, Taylor Valley (Cartwright and Harris, 1981). DVDP geological data established that any deep groundwater connectivity in Wright Valley would be through fractured bedrock (Ferrar Dolerite) and not sediments, as in Taylor Valley.

The formation of spatial extent of Wright Valley brines are relevant to Antarctic hydrological and geochemical processes, including those at subglacial and submarine interfaces (Foley et al., 2019), as well as hydrogeological processes on other icy planets (Toner et al., 2022). Furthermore, these brines may be a refuge for unique microbial life (Campen et al., 2019). Our

spatial investigation of Wright Valley did not resolve the potential for valley-wide groundwater connectivity, but did confirm
the presence of unfrozen brine saturated regions in the subsurface, and importantly, highlights regions that deserve further investigation.

*Code availability.* R code to reproduce the figure in this manuscript is available at: github.com/hdugan/WrightValley_AEM

*Data availability.* Data for the three flight lines presented in this manuscript are available at: github.com/hdugan/WrightValley_AEM Data for the project is available via the US Antarctic Program Data Center: usap-dc.org/view/dataset/601373

*Author contributions.* HD led manuscript preparation. DG, EA, NF, TB oversaw data collection, data processing, inversion, and interpretation. All authors contributed to field campaigns, and intellectual development of the manuscript.

*Competing interests.* None

*Acknowledgements.* This work was funded by the National Science Foundation Office of Polar Programs Award 1643536, 1643687, 1643775, and 1644187. The thoughtful critiques of two reviewers significantly improved this manuscript. We would like to thank the many contractors
who supported this work based out of McMurdo Station, with special thanks to the pilots and helicopter technicians that made this research possible.

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

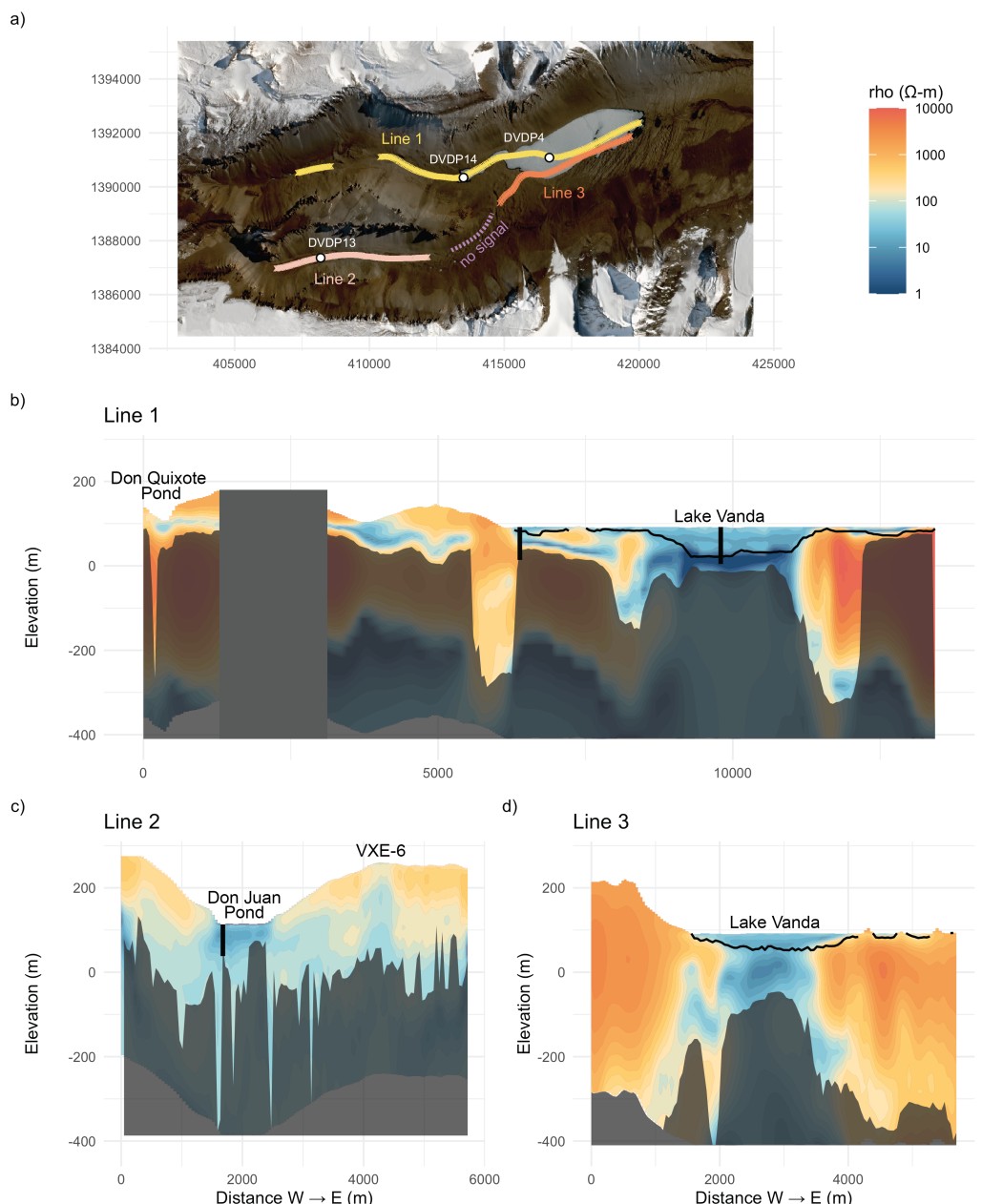

**Figure 1.** a) AEM flight lines over the North Fork of Wright Valley and Lake Vanda (Line 1), over Don Juan Pond (Line 2), and Lake Vanda towards the South Fork (Line 3). White circles show the location of the DVDP boreholes. Coordinates are in UTM-58S. Satellite imagery from Copernicus Sentinel data 2020, processed by ESA. b-d) Resistivity profiles of the three flight lines. The black line represents the sediment/water interface (bathymetry) of the Lake Vanda basin. Lower data are shaded beyond the depth of investigation. Black vertical rectangles show the approximate location and depth of the DVDP boreholes crossed by Line 1 and 3.