# Peer review of "Brief communication: The hidden labyrinth: Deep groundwater in Wright Valley, Antarctica"

_The Cryosphere, 2022_

## Author Comment (AC1)

**Authors' Response to Reviews of**

**Brief communication: The hidden labyrinth: Deep groundwater in Wright Valley, Antarctica**

Hilary Dugan *et al.*
*Cryosphere*
* * *
**RC:** *Reviewers' Comment*,     AR: Authors' Response

**1.  Reviewer #2**

**RC:**  *I read the paper with interest and have just a few suggestions that the authors may or may not want to include depending on scope. I don't have expertise in the resistivity analysis, so I have no comments on that.*

**RC:**  *The article talks a lot about the connectivity between DJP and Vanda, but an equally intriguing question is the ultimate origin of the DJP brine. This is mentioned in passing, but worth emphasizing more. Harris and Cartwright hypothesized that the brine is ultimately sourced from beneath the East Antarctic Ice Sheet i.e. west of DJP and through the Labyrinth. Is there any evidence for this from the resistivity data? A related question is if the DJP brine extends beneath the 'rock glacier'. It looks like west of DJP there is a low resistivity band at depth extending under the 'rock glacier', and very low values at the extreme that look like a numerical artifact.*

**AR:**  Intriguing question indeed! From our data we can't draw any conclusions on the source of the brine. In all of our electromagnetic surveying west of Don Juan Pond, which includes ground-based measurements on top of the rock glacier, and one survey in the Labryinth (which was incredibly difficult given the terrain), we found highly resistive material in the top few hundred meters. However, it is possible, and perhaps even likely that the groundwater sourced from the East Antarctic Ice Sheet would be much deeper than this.

**AR:**  As noted in Foley et al. (2019) [https://www.mdpi.com/2306-5338/6/2/54/htm] "In contrast to West Antarctica, interior East Antarctica is predicted to have overall lower basal melting and is largely surround by zones of net freezing near the coasts where the ice is thin but surface temperatures are cold [33]. This 'confining ring' of cold-based ice sheet margin helps to cryoconcentrate groundwater before it reaches the coast (Foley Figure 3). Thus, a low flux, high concentration groundwater scenario may be representative of much of coastal East Antarctica".

**RC:**  *The DJP transect also shows an interesting, vertical low conductivity feature to the E of DJP. E of DJP the elevation along the valley floor rises and then plateaus along a series of small basins. The first basin you encounter holds VXE-6 pond, which is typically dry at the surface but shallow groundwater occurs. This pond has a high CaCl2 content like Lake Vanda, but also high nitrate indicating considerable surface inputs. None of the other ponds have CaCl2. Cartwright and Harris analyzed this pond, and we recently analyzed it in Toner et al. 2022 (also discusses the mixing between NO3-rich and CaCl2-rich endmembers). I suspect that wind alone can't explain the CaCl2 in this pond; otherwise, why aren't other ponds similarly enriched? The resistivity data seems to suggest a connection between DJP and VXE-6, which would make sense. This would also put the DJP brine on the right path to connecting Lake Vanda, although the data can't show this. Too bad the flight line didn't extend to Lake Vanda!*

AR:     This is fascinating, and we agree that the pond chemistry aligns well with the AEM survey east of DJP. The flight line and additional ground-based surveys did cover the a South Fork transect between Lake Vanda and DJP, but no signal was picked up. So any connection must be deeper than our penetration depth (appox. 500 m).

RC:     *We recently published a paper on DJP and surrounding soils and groundwaters (https://www.sciencedirect.com/science/article/abs/pii/S0012821X22002187). One of the findings of the paper was that CaCl2 brine/salts like DJP infuse the Dolerite bedrock up to 200 m above the pond surface. The argument is that salt composition of the dolerite bedrock is so DJP like and different from surrounding soils, that inputs from wind alone can't explain the chemistry (you'd get mixing from nitrate-rich soils if deposited from wind), it must be primary. This supports a much stronger association between the DJP brine and the Ferrar Dolerite than previously thought. This suggests that you might "follow the Dolerite" to understand where the DJP groundwater is going. Might be interesting to include discussion about where the Dolerite is going, perhaps inferred from the strike/dip of the unit.*

AR:     It's a great paper and nicely timed to support our results. Thank you for pointing out all of the connections. We will add more discussion on the potential connection between hydrogeological pathways and the Ferrar Dolerite. Unfortunately, we can't say much about the subsurface geology beyond the McKelvey and Webb paper cited in Toner et al. 2022.

RC:     *Line 90: The conductivity of salt solutions depends on concentration and composition, and the conductivity decreases at very high concentrations for CaCl2. Could the low conductivity be explained in this way? Also, is the conductivity of CaCl2 different from equivalent ionic strength NaCl solutions. Would the porosity of the sediments and groundwater affect the result? Just wondering if the relatively low conductivity in DJP could be explained more easily.*

AR:     Correct, the conductivity of $CaCl_2$ is greater than the conductivity of an equivalent mass of NaCl. See https://www.researchgate.net/publication/280325585. Also for $CaCl_2$, the relationship between concentration (g/L) and specific conductance (mS/cm), is parabolic, with a maximum SpC of approx. 200 mS/cm at 300 g/L. Beyond 300 g/L, specific conductance decreases [see https://escholarship.org/content/qt5v01s3c6/qt5v01s3c6.pdf].

AR:     These two properties combined are relevant for thinking about the absolute relationship between inferred conductivity and brine composition. However, I believe the explanation of 1 ohm-m over DJP is still best explained by realizing that the top bin is a combination of a thin brine pool overlaying brine saturated sediments. As shown in Figure 2 from Mikucki et al (2017) [https://www.nature.com/articles/ncomms7831], brine saturated sediments would result in a higher resitivity than pure brine. This was applied to the calculation of brine porosity in Dugan et al. (2014) [https://agupubs-onlinelibrary-wiley-com/doi/full/10.1002/2014GL062431], where we found the resistivity minimum of 1.3 ohm-m indicated that only a fraction of the subsurface volume consists of the highly concentrated 0.15 ohm-m brine observed in the drill holes.

---

## Author Comment (AC2)

**Authors' Response to Reviews of**

**Brief communication: The hidden labyrinth: Deep groundwater in Wright Valley, Antarctica**

Hilary Dugan *et al.*
*Cryosphere*

**RC:** *Reviewers' Comment*,    AR: Authors' Response

**1. Reviewer #1**

**RC:** *This manuscript presents a fascinating first look at the electrical conductivity structure beneath Wright Valley, Antarctica, in order to evaluate long-standing predictions that subsurface brine conduits may link several water bodies in the valley. The work is well-argued, clearly presented, and nuanced in its analysis. The manuscript presents a compelling case that subsurface electrical conductivity anomalies are present east and west of Don Juan Pond, but that continuity between DJP and Lake Vanda could not be directly detected. High conductivity regions in the subsurface are clearly demonstrated in the North Fork of Wright Valley, which is strong evidence of conductive porewater solutions in the subsurface on that side of the Dias.*

**RC:** *One question raised by the paper is the role of sampling geometry in the interpretation of the SkyTEM results. Line 1 in Fig. 1 seems to show highly conductive material extending from Don Quixote pond in the west, nearly all the way to Lake Vanda in the east. The high conductivity horizon in the subsurface is interrupted by the data dropout and by a highly resistive block of material shortly after 5000 m in the along-track direction. Is there any morphological or topographic evidence that could suggest that the high conductivity region could extend continuously from DQP to LV, but that the region of continuity was simply not imaged in the footprint of Line 1? Line 1 seems to have been targeted to intersect with DVDP14 and 4, but is it possible that in doing so, subsurface, high conductivity materials to the north could have been missed? If so, it seems possible that a subsurface connection between salty solutions in pore spaces in North Fork do extend downslope all the way from DQP to LV. Likewise, is it possible that the sampling geometry of Line 3 is what causes the pinch out of the conductive zone in the subsurface west of LV? Some of this could be addressed by mentioning the cross-track sampling width of the SkyTEM.*

**AR:** The role of sampling geometry in data interpretation is certainly an important concern. It's easy to imagine brine conduits skirting the highly resistive regions that we surveyed. The cross-track sampling width is likely on the order of 60-70 m wide at the shallowest depths and extending to several hundred meters wide at the bottom of our profiles. A recent Groundbased vs Airborne paper has a few nice figs that illustrate this [https://doi.org/10.1016/j.coldregions.2022.103578] (Figs 5 and 6).

**AR:** Our flight lines were picked to align with the lowest elevation tracks of the valley, where one might assume brine would pool (although we know surface water and groundwater flow fields can be very different). From all of the flight lines, there are indications that there are highly resitive areas as shown in Line 1 and 3. In addition, a handful of ground-based surveys were conducted in 2017 at the far west end of Lake Vanda, as well as in the South Fork. The ground-based data aligns with the airborne data, suggesting that shallow along-valley conduits don't exist. However, we are the first to acknowledge that these data are just one piece of the puzzle and we can't definitively rule out the possibility of conduits that do not follow the valley floor.

AR:  For more technical information on data processing, you can find ground-based results from Lake Vanda in: https://www.sciencedirect.com/science/article/pii/S0165232X22000970, and a description of DJP processing in: https://academic.oup.com/gji/article/226/3/1574/6266462

AR:  In a revision, we will provide more information on flight line geometry and how it may impact interpretation

RC:  *Recognizing that the SkyTEM instrument is insensitive to shallow subsurface processes (i.e., one pixel for the upper 4 m of the soil/water column), the introductory text provides a somwhat facile or dismissive treatment of the role of surface and near-surface waters in affecting DJP chemistry and hydrology. For example, the text suggests that the variability in DJP extent and salinity indicates a hydrological driver beyond surface conditions. But surface conditions strongly control DJP lake level and extent as shown by (Dickson et al., 2013), who found a strong correlation between insolation (hence, snowmelt) and DJP spatial extent. Likewise, (Dickson et al., 2013) show input of water track solutions from the east, which also are associated with high insolation days which drive snowmelt and expansion of the active layer. (Hassinger and Mayewski, 1983) and (Dickson et al., 2013) both report that these near-surface water track solutions have high Ca, low Na, and excess Ca (Ca exceeding that which can be derived from dissolution of gypsum or calcite), which together, represent a potential contributing near-surface source for Ca-rich waters in DJP. The (Toner et al., 2017) modeling work is an important contribution to the understanding of potential subsurface processes in the DJP/Vanda region, but should not be considered an exhaustive analysis of hydrological contributors in the region because it largely considers only regional freshwater systems over the near-surface brines.*

AR:  We did not intend to be dismissive of surface hydrological inputs. As you point out, there has been a wealth of strong science that supports surface hydrology playing a role in DJP hydrochemistry. Our language stemmed from the aim of providing well-founded evidence (both empirical and modeling) of a deep groundwater interaction with DJP. Furthermore, we agree with Reviewer 2, that DJP brine chemistry cannot be explained from surface water alone.

AR:  In a revision, our aim would be to expand the literature review of DJP hydrology and provide more references to surface processes.

**Specific Comments:**

RC:  *Title: The (real) Labyrinth is a network of bedrock channels. And so, while I love the title, it seems like a network of bedrock and sedimentary fractures or pores in the subsurface really isn't what the manuscript suggests is occurring around DJP, DQP, and LV. In some ways this gets at my general comment above—there may very well be a hidden labyrinth of subsurface brine conduits—but can single TEM lines identify that geometry?*

AR:  I agree that we haven't been able to map a network of conduits in the subsurface, but the word labyrinth evokes a complicated maze-like network. The bedrock features on the surface are one such maze, but thus far it seems like the subsurface liquid pathways are still a complicated maze as well... even if we haven't fully solved the mystery. In a revision, we will be more clear about the possibility of "conduits" including the role fractured dolerite might play.

RC:  *Line 8. Are brine conduits implied by the observations or brine presence? I'd interpret "conduits" to mean localized zones of high permeability, which does not seem to be implied by the observations.*

AR:  Agreed that conduits implies a channel for conveying fluid. This is a important distinction from, say, trapped/isolate brine. We will be more specific in our language when refering to potential "conduits".

RC:  *Line 49. Suggest removing the editorial tone of "convincing arguments." It is a really excellent and intriguing*

*paper, but a more neutral introduction might help readers weigh the different arguments about water sources for DJP.*

AR: This will be addressed in a revision.

RC: *Line 97. How do you interpret the abrupt stop to the high conductivity zone at depth between DQP and Vanda? Is it a bedrock spur? A cold/dry permafrost pocket? Or evidence of brine diverging off the sensor path (in which case, there really is evidence for a subsurface labyrinth!).*

AR: Unfortunately there's not evidence for brine diverging off the sensor path... unless it's going down. An educated guess would say it's changing geology (Ferrar Dolerite to granite), but that is based off evidence of brine flow through Ferrar Dolerite near Don Juan Pond.

RC: *Line 100. It's really interesting that the low resistivity regions east and west of DJP extent up higher than the modern lake level. That could provide evidence of a perched saline aquifer that provides the hydraulic head observed in the brief artesian discharge episodes from the DJP boreholes, and would suggest that the low-resistivity zones east and west of the pond are at least partially connected to the brine in the ponds. This would be a really important finding because it differs from the classic groundwater interpretation for DJP (which is also invoked in the Toner et al, 2017 paper), which invokes cyclic deep groundwater upwelling. Line 2 seems to show that there is brine adjacent to and higher than the lake, suggesting that DJP solutions may not be exlusively upwelling from deeper sources.*

AR: This is a fascinating aspect that we didn't fully consider in our original submission. This also may align with the Toner et al. 2022 evidence of groundwater beneath VXE-6 pond which is at a higher elevation than DJP.

**2. Reviewer #2 response to Reviewer #1**

RC: *"...should not be considered an exhaustive analysis of hydrological contributors in the region because it largely considers only regional freshwater systems over the near-surface brines": In the Toner et al. 2017 paper we did model the surface brine evolution, and furthermore we considered all surface waters in Wright Valley as candidates. None of these surface waters can evaporatively evolve to form a DJP brine. Our recent paper Toner et al. 2022 provides an even more comprehensive look at deep, near surface, and surface water compositions in the South Fork of Wright Valley. This more recent paper supports the unique chemistry of DJP.*

RC: *Regarding the comments on surface discharges into DJP, we observed groundwater discharging east of DJP in the field, just as in Dickson et al. 2013. See the timelapse of discharge events over a month in the supplementary part of Toner et al. 2022. However, we also sampled many of these groundwater outflows, even during active outflow events, and analyzed the chemistry (unpublished unfortunately). There is no hint of any surface water contribution; the samples are pure DJP groundwater. Furthermore, these outflows have no observed connectivity to water tracks east of DJP, they simply upwell at the eastern edge the DJP playa. In my opinion, these are just groundwater outflows.*

RC: *Finally, regarding the discharge events and their correlation with insolation and snowfall, the most direct correlation with DJP groundwater levels appears to be air pressure, which is the expected behavior for a confined aquifer. There is data from the DVDP 13 borehole (sorry, again unpublished) that measures air pressure and water levels in the borehole, showing a strong correlation. Harris and Cartwright presented an analysis of the same, although there are many transient features that remain a mystery. We know that surface waters are contributing to DJP from streams on the western end of DJP from the rock glacier, but*

*their influence on the chemistry is very slight (possibly, this might explain the small nitrate component of DJP).*

RC: *All this is to say that a deep groundwater interpretation for DJP presented in this paper is well supported by the evidence.*

AR: We agree that the aforementioned modeling studies support the deep groundwater interpretation, and based our introduction around these results. Thank you for your responses.

---

## Author Response (AR2)

**Authors' Response to Reviews of**

**Brief communication: The hidden labyrinth: Deep groundwater in Wright Valley, Antarctica**

Hilary Dugan *et al.*
*Cryosphere*
* * *
**RC:** *Reviewers' Comment*,    AR: Authors' Response

**1. Editor**

**RC:** *Please add a sentence at the end of your introduction (L64) stating the purpose of the paper or the science question to be investigated.*

**AR:** We have added the following paragraph to the end of the introduction: *The aforementioned studies in Wright Valley were limited in their spatial observations by the difficulty in carrying out investigations in a remote and protected area. This study provides the first integrative overview of subsurface brines in Wright Valley using non-destructive geophysical measurements. Our research goals were to map water distribution and hydrological connectivity in order to further our understanding of permafrost hydrogeology.*

**RC:** *L82: repetition of 'ground', try and rephrase*

**AR:** Replaced one usage of 'ground'.

**RC:** *L89: replace 'know' with 'acknowledge'*

**AR:** Replaced

**RC:** *Add a sentence at the end of the discussion indicating the wider signficance of your results: the readership of The Cryosphere is wide, so please tell them why they should care about DV subsurface brines.*

**AR:** We have added a new sentence to indicate the importance and relevancy of this work.

**AR:** The conclusion now reads (excluding citations): *The formation of spatial extent of Wright Valley brines are relevant to Antarctic hydrological and geochemical processes, including those at subglacial and submarine interfaces, as well as hydrogeological processes on other icy planets. Furthermore, these brines may be a refuge for unique microbial life. Our spatial investigation of Wright Valley did not resolve the potential for valley-wide groundwater connectivity, but did confirm the presence of unfrozen brine saturated regions in the subsurface, and importantly, highlights regions that deserve further investigation.*